# Vulnerability of Prey Species to Predation by Two Sympatric Accipitrine Hawks in Rural and Peri-Urban Landscapes in Southern Finland

**DOI:** 10.3390/ani15040512

**Published:** 2025-02-11

**Authors:** Tapio Solonen

**Affiliations:** Luontotutkimus Solonen Oy, Neitsytsaarentie 7b B 147, FI-00960 Helsinki, Finland; tapio.solonen@pp.inet.fi

**Keywords:** behaviour, body mass, colouration, conspicuousness, habitat, population density

## Abstract

Vulnerability of prey species to predation may vary due to various intrinsic and extrinsic factors. Such factors also affect consistency of vulnerability between spatially and temporally varying environmental conditions. This study deals with consistency of vulnerability of prey species between two sympatric predators (the Eurasian sparrowhawk *Accipiter nisus* and the Eurasian goshawk *Astur gentilis*) and two landscapes. In rural landscapes, vulnerability of sparrowhawk prey increased with increasing body mass and decreased with increasing population density. In peri-urban areas, increasing population density and concentrated distribution reduced, and conspicuous habits and behaviour increased vulnerability of prey species. For the goshawk predation, vulnerability increased with increasing body mass of prey species consistently in both landscapes. These relationships suggest that the predator species encountered different challenges in different landscapes. There were clear differences in vulnerability of prey to predation between closely related sympatric predator species.

## 1. Introduction

Predation affects in various ways the mortality and regulation of prey populations [1,2,3]. Depending on the breadth of its diet, a predator can be characterised as a generalist or a specialist [4,5]. Generalist predators opportunistically prey upon all suitable prey items encountered, while specialists more or less select (or seem to select [6]) from available alternatives, showing (real or seeming [6]) preference for certain kinds of prey. In both cases, vulnerability of a prey species to predation by a predator (the risk of death from predation) can be characterised by the difference between the proportion of the prey species in the sample of prey items caught and the proportion of the species in the local pool of suitable prey [7,8,9].

Vulnerability of prey species to predation commonly decreases with increasing population density [9,10]. Density is negatively correlated with body mass as well [11,12]. Thus, vulnerability of prey species should increase with body mass to some extent [9]. Besides population density and body mass, vulnerability may also depend on various other body characteristics (such as colouration) and behaviour of prey species, as well as on characteristics of habitats and variable weather conditions [6,7,9,10,13,14,15]. Vulnerability of any one species depends not only on the species’ own characteristics and numbers, but also on the number of other prey in the same area, and it will vary locally [16].

The predator species dealt with in the present study are the Eurasian sparrowhawk *Accipiter nisus* (later sparrowhawk) and the Eurasian goshawk *Astur gentilis* (later goshawk), the most widespread and abundant avian predators of fully grown birds in Europe [17,18]. They exploit potentially almost all species of suitable size in the bird community of their breeding range [16,19]. The abundance of prey species can be estimated reliably in the field [20] and their vulnerability can be assessed from prey remains of the hawks [16,19].

I assessed how the predation by the sparrowhawk and the goshawk may affect the composition of prey communities in rural and peri-urban landscapes in southern Finland. The primary questions were as follows: (1) How consistent the vulnerability of prey species to predation is between different predator species and landscapes? (2) Which characteristics of prey species affect their vulnerability? Based on the above, I presumed that high density (a measure of abundance) decrease [9,10], intermediate body mass [16,19], and conspicuous colouration (brightness) [7] and ways of life (habits/behaviour) [10], as well as monotonous habitats lacking in variety (e.g., urban vs. rural, or open vs. forested), [10] increase vulnerability of prey.

## 2. Materials and Methods

### 2.1. Study Areas

The data on diets of the sparrowhawk and the goshawk came from two distinct areas in Uusimaa, near the southern coast of Finland (around 60° N, 25° E). The western area was the rural Inkoo–Lohja–Kirkkonummi district and the eastern one the surroundings of the Helsinki metropolitan area. Sampling covered approximately 500 km^2^ in both districts. However, study areas were not strictly limited sites but rather districts that served as common proxies of landscapes. The rural area was characterised by quite wide fertile forests and fields and sparse human habitation. In the peri-urban metropolitan area forests and fields were relatively small patches and their surroundings were densely populated built-up areas. The average density of human population was about 50/km^2^ in the rural landscape and around 2000/km^2^ in the peri-urban landscape (https://en.wikipedia.org/wiki/, accessed on 28 January 2025). In both districts, general densities for the sparrowhawk were around 10 pairs/km^2^ and for the goshawk around 5 pairs/km^2^ [21].

### 2.2. Prey Data

Prey remains were systematically collected near nests of the predators studied [22,23]. From the rural area, the data included 34 sparrowhawk nestings and 2424 prey individuals in 1979–1986 (Marcus Wikman, unpubl.), and 180 goshawk nestings and 1697 prey individuals in 1969–1983 [24,25]. From the peri-urban area, the data included 22 sparrowhawk nestings and 902 prey individuals in 1989–1990 [22], and 128 goshawk nestings and 2444 prey individuals in 2009–2016 [23]. For the present study, only the pooled data for both species and areas were available (Appendix A). Prey species which do not breed in the study areas (such as passing migrants) were excluded from the analyses, because their availability could not be estimated. Availability of prey species was characterised by population density and their suitability by body mass. Because prey remains were collected from plucking posts and nests, they represented mainly the prey brought by males. Due to the smaller size of males than females [16,19], the prey data were emphasised towards the lighter end of the prey size range of the species. Goshawks preyed on average approximately 10 times heavier prey than sparrowhawks, and the (peri)urban populations preyed on average lighter prey than the rural ones, the mean prey mass being 27.3/37.6 g and 312.2/366.0 g for urban and rural sparrowhawks and goshawks, respectively.

I calculated the expected number of prey by using information on abundance of prey relative to the breeding density of birds derived from the Finnish nation-wide line transect census [26] (the Finnish Museum of Natural History). The line transect results were transformed into relative densities of species [27]. The density estimates were derived from a total of 1071.9 km of line transects: 261.5 km in the western Uusimaa district, and 800.4 km in the Helsinki district. The density estimates used based on line transects that originated approximately (but not exactly) from those locations and time periods (within the area and time range) where and when the prey samples were collected.

### 2.3. Vulnerability to Predation

When estimating prey species’ vulnerability to predation, I assumed that the effects of detectability of prey species were similar for the predator and the observer. I estimated a logarithmic index of prey vulnerability as the observed log_10_-transformed relative frequency of prey items found at nests minus the log_10_-transformed expected number of prey based on the bird density estimates of the same area, adding a constant of 0.01 to avoid values of zero:Prey vulnerability index = log_10_ (proportion of prey + 0.01) − log_10_ (proportion of potential prey in the breeding bird community + 0.01).

A similar index of prey vulnerability has been used in various studies [7,8,9,28]. The expected number of prey relative to their abundance was estimated as the proportion of prey individuals of each species from the abundance based on censuses multiplied by the total number of prey individuals. Thus, an index value of zero implies that a prey species is consumed as expected based on its abundance, while an index value of +1 implies that a prey species is consumed 10 times more frequently than expected from its abundance, and an index value of −1 implies consumption 10 times less frequently than expected.

### 2.4. Statistics

In the analyses, I used linear regressions, an orthogonal (Deming) regression model [29] and generalised linear models (glm) [30], and visualised the results using the Grammar of Graphics (ggplot2) [31] in R (version 4.3.3, 29 February 2024) [32]. When examining factors that might affect vulnerability of prey to predation (response variable), I used population densities and body masses of species as primary predictor variables. Population densities were derived as described above, and body masses of species were derived from literature [33,34]. In addition, I included a set of subjective categorical variables broadly characterising differences in prey landscape (rural vs. peri-urban/urban), habitat (open (field)/intermediate (edge)/closed (forested)), distribution (even/intermediate (semi-colonial)/clustered (colonial)), colouration (dark/cryptic/spotted/light), and habits/behaviour (skulking/intermediate/conspicuous) based on my own observations and handbook data [17].

## 3. Results

The mean vulnerability score for the sparrowhawk prey (Appendix A) was 0.019 (±0.257 SD) (60 species) in the rural landscape and −0.025 (±0.191 SD) (47 species) in the peri-urban area. The corresponding values for the goshawk prey (Appendix A) were −0.016 (±0.530 SD) (41 species) and −0.012 (±0.432 SD) (48 species). There were 15–41 prey species shared between predators in different combinations of rural and peri-urban data sets (Appendix A). The shared prey species that occurred in both study areas included, among others, various thrushes *Turdus* sp., Eurasian jays *Garrulus glandarius*, and hazel grouses *Tetrastes bonasia*, which were numerous particularly in the goshawk diet, as well as chaffinches *Fringilla coelebs*, great tits *Parus major*, and tree pipits *Anthus trivialis*, which were numerous in the sparrowhawk diet.

Vulnerability of shared prey species to sparrowhawk predation in the rural landscape was significantly positively related to vulnerability to sparrowhawk predation in the peri-urban area (*r* = 0.623, df = 39, *p* < 0.001, two-tailed) (Figure 1). The average difference in vulnerability to sparrowhawk predation between these two districts was 0.027 (±0.032 SE) (*t* = 0.85, df = 40, *p* = 0.398, two-tailed).

Vulnerability of shared prey species to goshawk predation in the rural landscape was significantly positively related to vulnerability to goshawk predation in the peri-urban area (*r* = 0.623, df = 39, *p* < 0.001, two-tailed) (Figure 2). The average difference in vulnerability to goshawk predation between these two districts was 0.077 (±0.067 SE) (*t* = 1.15, df = 31, *p* = 0.259, two-tailed).

Vulnerability of shared prey species to sparrowhawk predation in the rural landscape was not significantly related to vulnerability to goshawk predation in the same district (*r* = 0.286, df = 25, *p* = 0.149, two-tailed) (Figure 3). The average difference in vulnerability to predation between sparrowhawk and goshawk prey was statistically significant (0.226 (±0.106 SE), *t* = 2.13, df = 26, *p* = 0.043, two-tailed), vulnerability being higher in sparrowhawk prey.

Vulnerability of shared prey species to sparrowhawk predation in the peri-urban landscape was significantly negatively related to vulnerability to goshawk predation in the same district (*r* = −0.513, df = 17, *p* = 0.025, two-tailed) (Figure 4). The average difference in vulnerability to predation between sparrowhawk and goshawk was not statistically significant (0.289 (±0.143 SE), *t* = 2.02, df = 18, *p* = 0.060, two-tailed).

Effects of various characteristics of prey species (Appendix A) on their vulnerability to predation varied to some extent between predator species and landscapes (Table 1, Table 2, Table 3 and Table 4). In particular, this was the case for sparrowhawk predation between rural and peri-urban landscapes (Table 1 and Table 2). For vulnerability to sparrowhawk predation, the effect of prey body mass was minor particularly in the peri-urban area. Vulnerability to sparrowhawk predation in rural landscapes was affected negatively by population density and positively by prey body mass (Table 1). In the peri-urban area, population density and distribution pattern affected negatively, and conspicuous habits/behaviour positively the vulnerability of prey species (Table 2). For goshawk predation, the positive effect of prey body mass was decisive in both landscapes (Table 3 and Table 4). In each case, the best models (based on Akaike Information Criterion (AIC)) included largely significant variables.

## 4. Discussion

This study aimed to show how consistently prey species are vulnerable to sparrowhawk and goshawk predation in two landscapes, and which characteristics of prey species affect their vulnerability. Based on various earlier studies (introduction), I presumed that high density decrease, and increasing body mass, bright colouration, conspicuous ways of life, and monotonous habitats increase vulnerability of prey.

### 4.1. Consistency in Vulnerability

Vulnerability of prey species to predation showed a considerable consistency between landscapes, although the data came largely from different periods of time (different years). This suggests that consistency in vulnerability was also temporal [9]. Stable environmental conditions and opportunistic predation can lead to consistency in vulnerability, while large-scale environmental changes and selective predation can cause changes in vulnerability. In an earlier study, differences among prey species in vulnerability to predation by the sparrowhawk and the goshawk were consistent between an agricultural area in Denmark and boreal forests in Finland, despite habitat structure and prey and predator densities differing considerably among landscapes [8]. Differences in prey were probably due to pronounced differences in habitats. There may be marked seasonal changes also in prey and habitat use of predators [10,35].

The difference in prey vulnerability between the two predators (vulnerability being higher in sparrowhawk prey) in rural landscape might be due to the higher agility and consequent greater predation efficiency of the sparrowhawk in densely forest-dominated home range. The negative association in prey vulnerability between the sparrowhawk and the goshawk in urban habitat might be due to a better availability of smaller-sized species in the local prey pool compared to the situation in the rural landscape. These ideas require, however, further examination.

### 4.2. Body Mass and Population Density

In accordance with several earlier studies [10,13,36], the present study highlights the most prominent impact of prey body mass and population density on vulnerability to predation. For sparrowhawk predation, however, the effect of prey body mass was minor. This seemed to be a quite general rule [6]. However, prey species with larger body mass are more vulnerable to sparrowhawk predation in harder environmental conditions of northern Finland [7]. As expected, within the size range of suitable prey, vulnerability to predation was highest in prey species of intermediate body mass [10,13,16,19,22]. Larger prey might be selected for larger body size and/or larger individuals might be more selective when hunting large prey [37].

Predator size determines prey size to a considerable extent [38,39]. In the present case, the body mass of the larger predator (the goshawk) was five- to sixfold that of the smaller one (the sparrowhawk) [33]. This largely explains the predator-specific differences in vulnerability of prey species. The suitability of prey depends not only on its energetic profitability but also on its accessibility (catchability) and manageability [13,16]. Predators are more efficient at handling prey of intermediate body size because it takes a relatively longer time to manipulate small prey and capture large prey [16,19]. The relationship between vulnerability to predation and body mass may be complicated if larger prey species are considered less likely to die from predation due to their ability to deter predators [40].

Contrary to a larger-scale study [9], the present results suggest that prey population density was not an important factor affecting vulnerability of prey to goshawk predation. In line with various other studies, however, present results show that vulnerability of prey to sparrowhawk predation decreases with increasing prey density [6,8,10,41]. Thus, prey individuals living in dense populations were less vulnerable to sparrowhawk predation than those in less dense populations. Densities of small-sized species are, in general, higher than those of larger species [12,42]. High abundance of certain prey maintaining predators at a relatively high density may increase the vulnerability of other species [43].

### 4.3. Conspicuousness: Colouration and Behaviour

Conspicuousness of prey species is affected by many independent factors, but the interpretation of their effects may be complicated [10,44,45]. Conspicuous colours may increase predation risk [45,46,47]. An obvious determinant of conspicuousness is brightness of colours. However, the effects of background and behaviour must be taken into account. Cryptic species are difficult to detect by predators if their colouration is similar to that of the background [46,47,48].

Prey plumage brightness has shown to be the most important factor determining vulnerability to sparrowhawk predation [7]. In adult prey individuals, male brightness may be more important than female brightness in explaining prey vulnerability [7,14,44]. Predators may attack cryptic females similarly [14] or more often than conspicuous males [44]. Prey abundance does not affect the relationship between predation vulnerability and plumage brightness, because both rare and common species with bright plumage suffer higher predation [7]. Those species that seemed to be the most vulnerable ones have also been relatively conspicuous in the field.

Some species that lead a skulking way of life, even though abundant in the field, have been rare in the prey of the sparrowhawk [22]. Vulnerability may decrease with increasing foraging height and perch height, and increase with mean exposure of prey species [10]. Differences in the habitat use of prey species and the hunting behaviour of the predator are likely decisive factors modifying the prey composition of a predator [16,49,50]. The hunting technique of predators is closely related to the vulnerability of potential prey species [51,52,53]. Vulnerability of prey to sparrowhawk predation seems to be affected by foraging habits of the predator as well as to characteristics of the habitat and those of the prey species itself [6]. The large variability of vulnerability indices in many species suggests that various species-specific characteristics were not, in general, important determinants of predation risk.

### 4.4. Methodological Considerations

Besides the fact that the differences in vulnerability may be real and caused by differences in catchability due to structural or behavioural differences between species, results may be biased by various kinds of methodological shortcomings, concerning data on both available and used prey. In spatially and temporally large-scale studies like the present one, the gathering of data may be both laborious and expensive. Therefore, it commonly relies on only the best data available. As in many other related studies [8,9,13,36], the density estimates used here were not exactly from those locations and time periods where and when the prey samples were collected. However, it is well known that the abundance of prey species may vary between years.

The scarcity of significant relationships between the subjective categorical variables and vulnerability may be at least partly due to various undetected characteristics that were probably included in the broad categories used. For instance, the colouration of prey may vary according to species, also after sex and age. In addition, in different cases colouration may signal different things for predators [44,46,47,54]. Estimates of the abundance of available prey and those of prey taken by predators may be variously biased due to differences in the conspicuousness of prey species in the field or in remnant samples [16]. In general, the bias in prey remains is well acknowledged [22,55,56], with larger-sized and brightly coloured species overrepresented and small and dull-coloured species underrepresented with respect to numbers in the environment [57,58].

Results may also be affected if samples of the available and consumed prey compared do not represent the same geographical areas adequately [6,15,22]. Important food sources of the predator may have been located outside the area monitored for the estimates of abundance of available prey. Thus, for instance, unevenly distributed forest bird species that, for one reason or another, concentrate in forest-like habitats near human settlements (parks, gardens, yards) and that are heavily exploited by hawks, may have been estimated to be more vulnerable than they really are. Such species were estimated to be especially vulnerable in the data sets of urban areas. Thus, conspicuously uneven distribution among habitats of some bird species, and similar exploitation patterns of hawks, may make the vulnerability estimates based on general densities of prey unreliable. Local, heavily exploited food resources (such as concentrations of unevenly distributed species) may reduce predation pressure on some other, evenly distributed species. Single individual predators can learn or specialise in using some locally tempting resources such as accessible broods or dense urban populations as a consequence of good hunting success [6,10,16,22].

## 5. Conclusions

Vulnerability of prey species to sparrowhawk and goshawk predation seems to be relatively consistent between predators and landscapes. Positive relationships between predators suggest that they prey largely independently from each other, while negative relationships suggest some interaction between predators. Higher vulnerability of shared prey species to sparrowhawk predation suggests that these species were less important prey or harder to catch for the goshawk. The present results highlight the prominent impact of prey body mass and population density on vulnerability to predation. Various species-specific characteristics do not seem to be important determinants of predation risk. A major difficulty in assessing vulnerability is the measuring of availability of prey in the real foraging area of the predator. Availability of prey has been measured on the basis of overall density estimates of prey species in the district concerned, and it has been assumed that predators forage evenly throughout the area. Thus, the effects of heterogeneity of the environment and uneven prey availability have been ignored.

## Figures and Tables

**Figure 1 animals-15-00512-f001:**
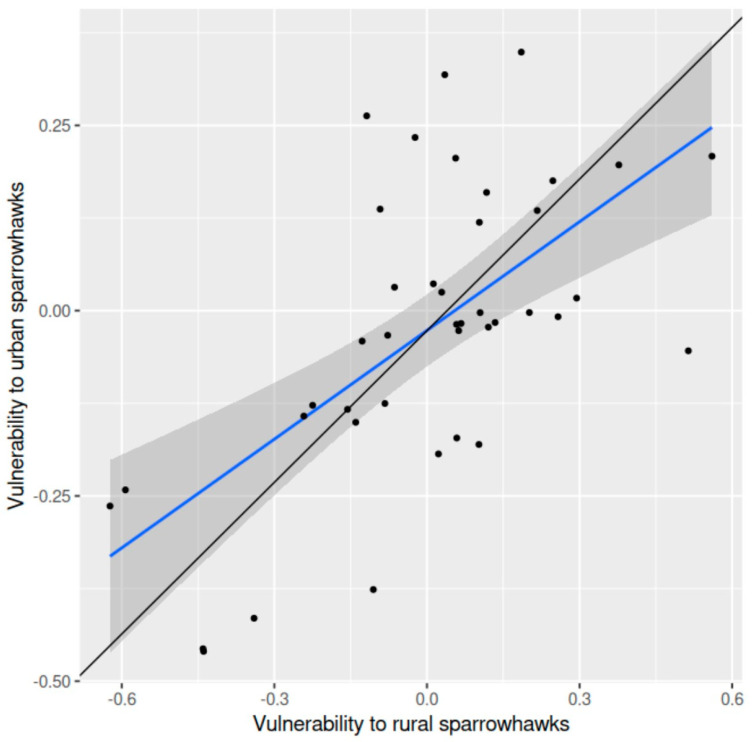
Vulnerability of shared prey species to sparrowhawk predation in a rural landscape in relation to vulnerability to sparrowhawk predation in a peri-urban area in southern Finland: the ordinary regression line (blue) fitted to the data (points), the 95% confidence band (grey) and the orthogonal regression line (black) (f(x) = 0.682x − 0.027; confidence intervals, slope 0.441–1.084, intercept −0.078–0.030) are shown.

**Figure 2 animals-15-00512-f002:**
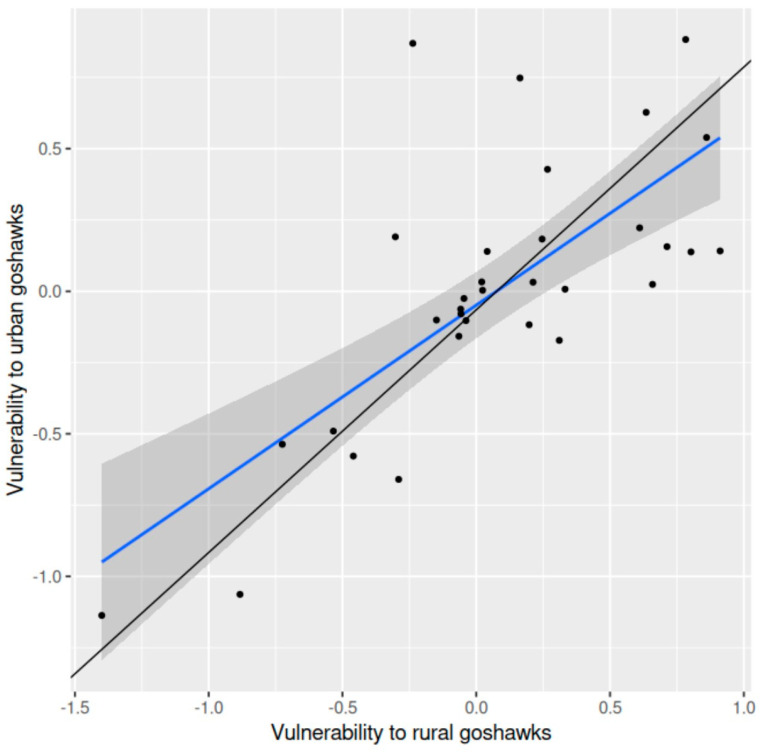
Vulnerability of shared prey species to goshawk predation in a rural landscape in relation to vulnerability to goshawk predation in a peri-urban area in southern Finland: theordinary regression line (blue) fitted to the data (points), the 95% confidence band (grey) and the orthogonal regression line (black) (f(x) = 0.851x − 0.065; confidence intervals, slope 0.604–1.143, intercept −0.163–0.058) are shown.

**Figure 3 animals-15-00512-f003:**
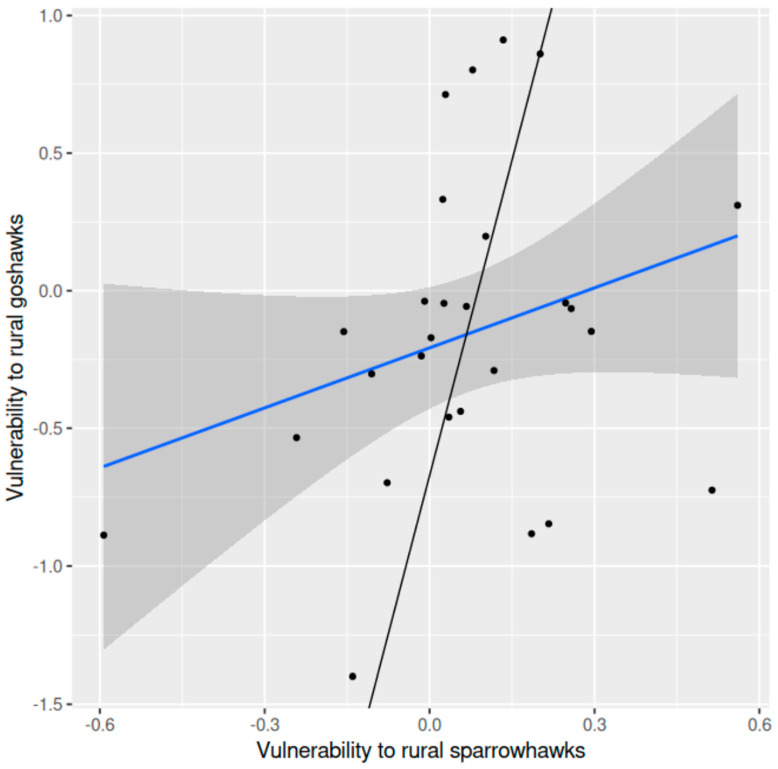
Vulnerability of shared prey species to sparrowhawk predation in relation to vulnerability to goshawk predation in a rural area in southern Finland: the ordinary regression line (blue) fitted to the data (points), the 95% confidence band (grey) and the orthogonal regression line (black) (f(x) = 7.643x − 0.672; confidence intervals, slope −48.347–63.433, intercept −6.131–4.753) are shown.

**Figure 4 animals-15-00512-f004:**
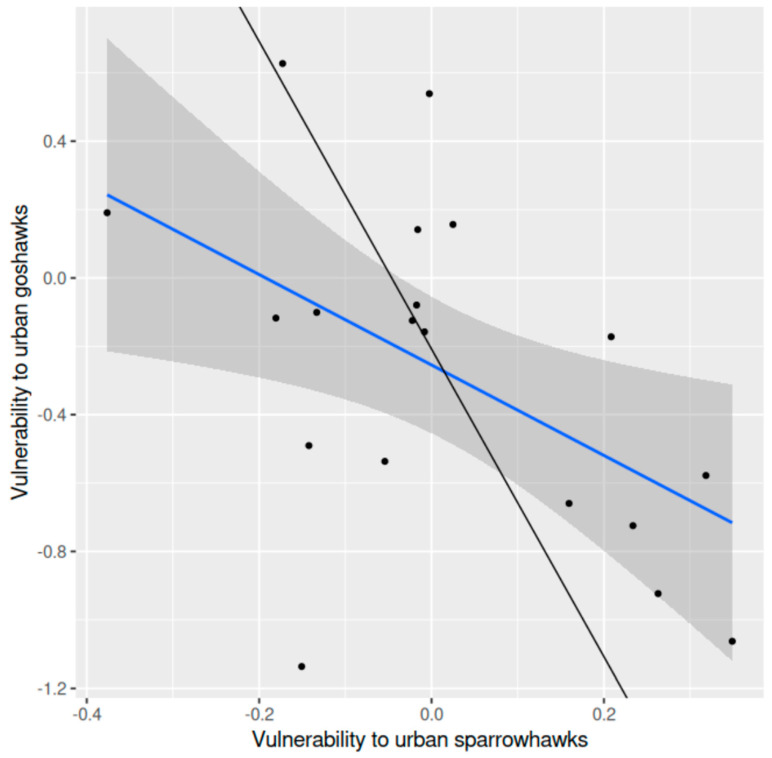
Vulnerability of shared prey species to sparrowhawk predation in relation to vulnerability to goshawk predation in a peri-urban area in southern Finland: the ordinary regression line (blue) fitted to the data (points), the 95% confidence band (grey) and the orthogonal regression line (black) (f(x) = −4.499x − 0.208; confidence intervals, slope –22.813––2.088, intercept −1.044–0.206) are shown.

**Table 1 animals-15-00512-t001:** Generalised linear models (glm) of the relationships between vulnerability of prey species to sparrowhawk predation in rural landscapes (response variable) and body mass (log_10_-transformed), population density (log_10_-transformed), spatial distribution, habitat, colouration, and conspicuousness of habits of prey species (predictor variables). Effects in bold are statistically significant. Adjusted R^2^ for the total model is 0.0726. AIC = Akaike Information Criterion.

Variable	Estimate	SE	*t*	*P*	r^2^
Total model AIC: 11.36					
(Intercept)	−0.4239	0.2982	−1.422	0.1610	
log body mass	0.1605	0.0943	1.702	0.0946	0.0866
log density	−0.0950	0.0453	−2.097	**0.0408**	**0.0883**
Distribution	−0.0180	0.0827	−0.218	0.8283	0.0006
Habitat	0.0388	0.0511	0.760	0.4509	0.0001
Colouration	0.0241	0.0622	0.387	0.7002	0.0028
Habits	0.0342	0.0466	0.734	0.4660	0.0051
Best model AIC: 5.08					
(Intercept)	−0.2018	0.1264	−1.597	0.1159	
log body mass	0.1530	0.0831	1.840	0.0709	
log density	−0.0759	0.0406	−1.871	0.0665	

**Table 2 animals-15-00512-t002:** Generalised linear model (glm) of the relationships between vulnerability of prey species to sparrowhawk predation in peri-urban landscapes (response variable) and body mass (log_10_-transformed), population density (log_10_-transformed), spatial distribution, habitat, colouration, and conspicuousness of habits of prey species (predictor variables). Effects in bold are statistically significant. Adjusted R^2^ for the total model is 0.2003.

Variable	Estimate	SE	*t*	*p*	r^2^
Total model AIC: −24.41					
(Intercept)	0.1909	0.2398	0.796	0.4307	
log body mass	−0.0076	0.0726	−0.105	0.9169	0.0016
log density	−0.0981	0.0440	−2.228	**0.0316**	**0.0591**
Distribution	−0.2067	0.0687	−3.009	**0.0045**	**0.0209**
Habitat	−0.0808	0.0501	−1.614	0.1143	0.0268
Colouration	0.0335	0.0515	0.650	0.5196	0.0191
Habits	0.0830	0.0346	2.395	**0.0214**	**0.0691**
Best model AIC: −27.13					
(Intercept)	−0.0059	0.0976	−0.061	0.9519	
log density	−0.1156	0.0413	−2.795	**0.0077**	
Distribution	−0.1603	0.0620	−2.585	**0.0132**	
Habits	0.0890	0.0321	2.770	**0.0082**	

**Table 3 animals-15-00512-t003:** Generalised linear model (glm) of the relationships between vulnerability of prey species to goshawk predation in rural landscapes (response variable) and body mass (log_10_-transformed), population density (log_10_-transformed), spatial distribution, habitat, colouration, and conspicuousness of habits of prey species (predictor variables). Effects in bold are statistically significant. Adjusted R^2^ for the total model is 0.4847.

Variable	Estimate	SE	*t*	*p*	r^2^
Total model AIC: 45.89					
(Intercept)	−0.9196	0.6262	−1.469	0.1511	
log body mass	0.6044	0.1621	3.728	**0.0007**	**0.4976**
log density	−0.1022	0.0966	−1.058	0.2974	0.2953
Distribution	−0.0973	0.1886	−0.516	0.6092	0.0145
Habitat	−0.0065	0.0994	−0.065	0.9485	0.0028
Colouration	0.0578	0.0934	0.618	0.5405	0.0487
Habits	−0.1275	0.0870	−1.466	0.1518	0.0626
Best model AIC: 40.66					
(Intercept)	−0.7538	0.3593	−2.098	0.0428	
log body mass	0.5355	0.1329	4.030	**0.0003**	
log density	−0.1190	0.0907	−1.313	0.1974	
Habits	−0.1379	0.0829	−1.664	0.1046	

**Table 4 animals-15-00512-t004:** Generalised linear model (glm) of the relationships between vulnerability of prey species to goshawk predation in peri-urban landscapes (response variable) and body mass (log_10_-transformed), population density (log_10_-transformed), spatial distribution, habitat, colouration, and conspicuousness of habits of prey species (predictor variables). Effects in bold are statistically significant. Adjusted R^2^ for the total model is 0.4313.

Variable	Estimate	SE	*t*	*p*	r^2^
Total model AIC: 40.86					
(Intercept)	−0.5960	0.4887	−1.219	0.2300	
log body mass	0.3857	0.1097	3.516	**0.0011**	**0.4440**
log density	−0.0036	0.0638	−0.057	0.9547	0.1358
Distribution	0.0981	0.1589	0.617	0.5408	0.1614
Habitat	−0.0713	0.1034	−0.690	0.4944	0.1086
Colouration	−0.0508	0.0693	−0.733	0.4680	0.0143
Habits	−0.0319	0.0789	−0.404	0.6882	0.0096
Best model AIC: 33.28					
(Intercept)	−0.9971	0.1725	−5.779	<0.0001	
log body mass	0.4577	0.0772	5.927	<**0.0001**	

## Data Availability

Data are available in Appendix A.

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
