# Peer review of "Vulnerability of Prey Species to Predation by Two Sympatric Accipitrine Hawks in Rural and Peri-Urban Landscapes in Southern Finland"

_animals, 2025, doi:10.3390/ani15040512_

Round 1
Reviewer 1 Report
Comments and Suggestions for Authors
Thank You very much for an opportunity to review this interesting and valuable paper. Nevertheless I have several suggestions with can improve the manuscript.
In line 46 there is information about negative correlation between density and body mass. It is generally true in natural conditions. Yet, in human transformed landscape relatively big prey might be also very abundant, i.e. domestic pigeons, poultry, pheasants released in hunting grounds etc.
Study area should be detailly described. I don’t see most of basic information: study areas size, GIS description of the landscape, statistically significant differences between selected habitats parameters. I would like to know what was the differences between peri-urban and rural study sites. Data about avian predators density is also missing. I suggest to add table with selected landscape parameters of rural and peri urban study sites.
Section ‘Prey data’ is also unsatisfactory. I suggest to add tables (maybe as appendix) with diet composition of goshawk and sparrowhawk. Data about diet composition were collected during very long time. In case of goshawk in rural area it was 15 years. Its also not clear what does it mean 180 goshawk nest? Its breeding attempts, 180 different nest in different territories or 180 territories… it should be clarified. If we take under consideration numbers of prey items, number of nests and study period, sample size related with one nest in one breeding season is very low. Another problem its calculation of prey density, as I understand it is rather density indicator which can be compared between different breeding seasons. I think that this data are only roughly relevant to different breeding territories in particular seasons. This attitude is one of the biggest disadvantage of this paper. As I understand also study sites changed significantly during very long study period 1969-2016, is should be discussed.
In line 315 some information about significant dimorphism of males and females should be highlighted. When we compare female of sparrowhawk and male of goshawk differences are not so big.
Information in lines 372-375 should be better discussed, bird recognized colors differently than humans. Which birds are better visible for goshawks and sparrowhawks is an open question.
I suggest to add some relevant papers about food composition of study species.
Gryz J., Krauze-Gryz D. 2018. Density dynamics, diet composition and productivity of sparrowhawk Accipiter nisus L. population in central Poland. Forest Research Papers 79(3): 245-251 https://doi10.2478/frp-2018-0024
Gryz J., Krauze-Gryz D. 2019. Pigeon and poultry breeders, friends or enemies of northern goshawk Accipiter gentilis? A long term study of the population in central Poland. Animals, 9(4), 141, https://doi.org/10.3390/ani9040141
Reviewer 2 Report
Comments and Suggestions for Authors
Prey type and frequency was studied in about 350 nests of two birds of prey (goshawk and sparrow hawk) over almost half a century (start: 1969, end: 2016). An unusually long and interesting period. The time span is sufficient to identify changes in the diet of two raptor species, whether due to changes in availability, age of the predators, or changes in the preferences or habits of individuals of each raptor species. However, the aim of the paper is not so much to study changes in the diet of raptors over time, but to investigate whether the composition of prey communities is the result of predation pressure (lines 19-21). This research has a laudable aim that goes beyond traditional studies of how food availability determines a predator's diet. Instead, it examines how the predator's diet influences the prey community (lines 19-21). The paper asks specific questions such as a) What are the differences in vulnerability between prey species and b) How do characteristics such as body size vary between prey found in different raptor nests? An advantage of this study is that it didn't require estimating prey availability, which is usually a major field task, because the information on prey was available. Admittedly, on a larger scale than the study areas, but this may have been appropriate for the research purposes. There were two zones that differed mainly in landscape structure: a peri-urban zone and a rural zone. The manuscript therefore structured its research on the prey community along two mental axes: differences caused by raptor species (goshawk vs. sparrow hawk) and those caused by landscape structure (rural vs. peri-urban).
The paper concludes that different challenges were found by different predator species in different landscapes (lines 16-17). While this is an expected conclusion, the details of the results were not. For example, the vulnerability of species in the diet of both predators (referred to in the manuscript as "shared prey species") was determined by the landscape of each study area (rural vs. peri-urban). It was also found that the vulnerability of shared prey species was greater in sparrow hawk nests than in goshawk nests. A third result was that the vulnerability of prey communities was determined by the size and density of prey in each study area. It is an interesting paper and certainly has valuable data that should be of use in future studies.
The paper has important methodological limitations that strongly influence the results. Some of these are well known to the author and I will not comment on them further as they are openly acknowledged in the Discussion (pp. 15-16). I will comment on other aspects which, in contrast to the main limitations accepted in the manuscript, can and should be improved. The first is a clearer definition of the data unit of statistical analysis. The sampling unit was the raptor nest (N=364 nests, lines 84-87) or its equivalent as a breeding area. The unit of statistical analysis was the prey species: 60, 47, 41 and 48 species (lines 132-134). The results (figures and statistical text) indicate between 17 and 39 degrees of freedom for each analysis, but this is not sufficient to understand the decline in numbers. An explanation in the methods needs to be added. This is a methodological aspect that is more important than it seems.
Also, the methods do not adequately explain how the "proportion of prey" (lines 108-109) is averaged between nests. Each nest gives a different proportion with a different confidence interval, depending on the total number of prey identified. For example, a proportion of 0.10 calculated on a total of N=10, N=20 and N=200 prey has confidence intervals (95%) of (0.02,0.40), (0.03,0.30) and (0.07,0.15) respectively. The mean number of total prey per nest was N=20.5 (7467 prey / 364 nests). However, the intervals in each of the 364 nests must have been very different. Differences between nests in the confidence intervals should not be ignored, as they should be propagated into the calculation of the mean of the proportions. Error propagation is easily explained in Harvard (2007). A recent example of the importance of error propagation is provided by Simmonds & Jones (2024). It is, or could be, important to recover the ranges of each species in each nest for the calculation of confidence intervals of the mean proportion.
In addition, the calculation of the mean proportion should be weighted by the total number of prey in each nest to avoid giving the same weight to nests with few prey and nests with many prey. Both statistical aspects, i.e. a) the error propagation of each proportion in the calculation of confidence intervals of the mean, and b) the weighting of each proportion in the calculation of the mean, are not minor issues. The first methodological change concerns whether or not to exclude a prey species in the manuscript. The second methodological change will affect the numerical value of the mean, depending on whether or not each data point is weighted by the number of prey items it contributed to the total. It may be impossible to recalculate if there is no access to field data, or unjustified if a shortcut is found to discard nests because of their low contribution. It may also be preferable to rebut this comment in the response letter rather than redo the calculations. These are all viable options in principle. In any case, further methodological explanation of the process of calculating the proportion of each prey item is essential before calculating the 'Prey Vulnerability Index'. Error propagation should also be calculated for this index, as it is the sum of two proportions, each with confidence intervals. As an aside, the figures show the mean vulnerability of each prey species, but without the confidence intervals, which means that the variability of the sampling unit (the nest or breeding area) has been excluded in the statistical analyses of the variables constructed from the sampling unit. This is an error that can be corrected by repeating the statistical analysis using the proportion of each nest as the unit of analysis. This increases the statistical sample size but also the variability. In any case, recalculating the results does not guarantee that the same conclusions will be reached.
Another aspect that could improve the work is the inclusion of prey species that did not breed in the study areas. Especially if their abundance among the identified remains was remarkable. This is relevant information for calculating the proportions of other prey species in the nests. The manuscript indicates that they were excluded from the calculations (lines 88-89), but does not explain why. Please forgive me if this has been explained and I have not understood it. In any case, both summaries should include this key information. For example, line 11 should read "and breeding in two landscapes". Line 20 should read "prey communities breeding in rural and peri-urban landscapes". And the methods should explain why prey species are excluded, which may also affect the proportion of other species consumed. This is a solvable problem, but it will mean recalculating all the results.
Another area for improvement is the way regressions are calculated in the figures when one variable has no effect on the other. Figures 2, 3, 4 and 5 show a regression line without any justification as to why there is a dependent variable (y-axis) caused by the independent variable (x-axis). The linear regression (page 4, line 118) should be calculated using an orthogonal regression model. See Wikipedia for a simple explanation: https://en.wikipedia.org/wiki/Deming_regression . In the main text, a correlation coefficient is calculated, which is the appropriate way to measure the association between two variables without an a priori cause-effect relationship (lines 143 and 153).
Generalized linear models are fitted, but they are not sufficiently explained. For example, Tables 1 to 4 show the final model with six variables. Most of them are not significant and therefore cannot be a final model but an initial model. The r-squared value of the overall model is not a sufficient criterion for model adequacy. For the model to be acceptable, in these cases some other standard statistic must be provided, such as the AIC.
Minor comments
Page 3, Figure 1. It may be more useful to increase the detail of the study area than to locate it within the country. The current map could be reduced and added into a sketch showing the extent of the two zones and the distance between them. What was the area of each study zone in square kilometres? In addition, it may be important to include prey availability transects in each district where the study areas are located to visually assess the overlap between nest distribution and transect distribution.
Pages 12-13. The columns in Tables 1-4 are not correctly aligned.
Pages 16-18. References. There are many gaps between words, perhaps to force a fit on both sides of the page. But in some cases there is a split word. See, for example, lines 436 ("re lation") and 513 ("im ages"). I am aware that these minor editorial errors will be dealt with by the technical editors if your manuscript is accepted for publication. So please bear with my nitpicking.
References
Warning: These references are provided solely in support of the reviewer's comments. Please do not include them in the updated manuscript.
Harvard. (2007). A summary of error propagation. Harvard University Press. Retrieved September 2022 from https://www.academia.edu/14672973
Simmonds, E. G., & Jones, O. R. (2024). Uncertainty propagation in matrix population models: Gaps, importance and guidelines. Methods in Ecology and Evolution, 427-438. https://doi.org/10.1111/2041-210x.14276
Reviewer 3 Report
Comments and Suggestions for Authors
Review of the article titled “Vulnerability of Prey Species to Predation by two Sympatric Accipitrine Hawks in Rural and Peri-urban Landscapes in Southern Finland” (ID animals-3427333).
Solonen et al. present a survey on the impact of raptor predation on the ulnerability of bird prey considering raptor species size (large goshawk and small sparrowhawk) and landscape (urban vs periurban).
First, it is impressive work concerning sampling and it is not important that these samples were collected a year ago. It would be a great topic to repeat such a study in current conditions. I found this study interesting and an important contribution to the ecology of birds of prey. I thought that based on such kind of data nothing new could be found in this matter but the method used in this study shows new findings.
I do not have any comments or questions about to study design except that nothing is informed about bird species taken by these raptors. I understand that the Author intended to draw some generalities but it would be good also to know some prey species-specific details (only some are in l. 136-140).
I wonder about excluded species that did not breed in the study area – these were migrants? In the paper I failed to find any raw data e.g. in supplementary and such information should be added, from predators and transect censuses.
What do you mean by “…, although the data came largely from different periods of time.” (l. 290) – I supposed that all prey data were from breeding periods. Line transect data were from which years (it is not specified in Methods)?
I do not think that fig. 1 is important and should be removed.
Some plots should be re-drawn to exclude parts with absent data (e.g. below -0.75 and above +0.75 in fig. 2, the same for axis Y and in the next figures).
Reviewer 4 Report
Comments and Suggestions for Authors
This study used avian prey remains at predator nests of two species to determine whether prey vulnerability is similar between different habitats and between predators. In addition, the study assessed the relevance of various prey traits including density and body mass in explaining prey vulnerability. The study follows in the footsteps of similar studies conducted in Finland and adds to our knowledge of factors influencing prey vulnerability by contrasting two habitats at about the same time period. The author has acknowledged clearly the limits of such studies. My comments below aimed at clarifying some methodological issues.
Line 45 : Because it is the central theme of the paper, a definition of vulnerability to predation is needed. Does it entail the risk of attacks by predators and/or the risk of death from predation?
Line 47 : The relationship between vulnerability to predation and body mass is more complicated because larger prey species are considered less likely to die from predation due to their ability to deter predators. A recent review of the ecological factors that affect susceptibility to predation is provided in Birds (2023, volume 4, pages 73-84).
Line 66: Earlier in the introduction, high population density was said to decrease predation pressure. Also, there is a distinction between abundance and density so the labels should not be mixed up. May I suggest a table to highlights the predictions along with more specific references? I think it would be important to develop these predictions. It was not clear to me how colouration is related to vulnerabilty to predation and how it is measured. Same goes for habitat type. What does it mean when a habitat is monotonous? Rural and peri-urban can conceivably differ in many ways including predator and prey density. Ways of life can mean anything. Can you be more specific? I am thinking especially about group living as it has been shown to influence risk of predation. I note that the introduction is rather succinct. There is room to develop these ideas here.
Figure 1 : Is it possible to zoom in in the two study area to get more details?
Line 87 : I note that the two areas were not examined at the same time period. Environments change from year to year. Might be important to discuss possible consequences of a year gap between the two habitats.
Line 127 : Does colonial mean the species is a colonial breeder or does it mean that the prey species is found in groups? Is there a justification for using four classes for colouration? Surely, there must be cases where a species is dark as well as spotted or cryptic. Please explain how such classes are defined. The same can be said for distribution and habits. I know these European species well and I am not sure I would be able to fit them so snugly into these different classes.
Line 128 : What does pronounced mean in the context of habits?
Line 145 : A paired t-test would be appropriate here given that vulnerability was assessed for shared prey in the two habitats, no?
Table 1 : We know that the effect of prey body mass is not linear. May I suggest to use a polynomial term in the multi-variable model? Same for the goshawk data.
Line 293 : But the reference listed here found clear evidence for consistency at least spatially.
Line 301 : These sentences are not clear. Please expand.
Line 326 : The effect of density was obtained controlling for the effect of body mass. So it appears that this effect is independent of body mass.
Line 356 : This is a good section on the limits of such studies. Perhaps two prey traits that have not been considered here are group size and whether the species forages on the ground or in trees. But the point remains that many traits are important when choosing prey species.
Line 394 : There was little discussion about the negative association between prey vulnerability between the two species in the urban habitat as well as the difference in prey vulnerability between the two species. This could be extended.
Round 2
Reviewer 1 Report
Comments and Suggestions for Authors
This is my second revision of this paper. I have no further remarks and I am satisfied with corrections.
Reviewer 2 Report
Comments and Suggestions for Authors
Most of the concerns raised in the previous peer review have been addressed to some extent. I have no further comments to improve the manuscript.